# Brachytherapy in the Treatment of Soft-Tissue Sarcomas of the Extremities—A Current Concept and Systematic Review of the Literature

**DOI:** 10.3390/cancers15041133

**Published:** 2023-02-10

**Authors:** Johannes Neugebauer, Philipp Blum, Alexander Keiler, Markus Süß, Markus Neubauer, Lukas Moser, Dietmar Dammerer

**Affiliations:** 1Department of Orthopaedics and Traumatology, Krems University Hospital, 3500 Krems, Austria; 2Karl Landsteiner University of Health Sciences, 3500 Krems, Austria; 3Department of Orthopaedics and Traumatology, Medical University of Innsbruck, 6020 Innsbruck, Austria; 4Department of Trauma Surgery, BG Trauma Center Murnau, 82418 Murnau, Germany

**Keywords:** high-dose brachytherapy, high-grade soft-tissue sarcoma, intraoperative brachytherapy, soft-tissue sarcoma of the extremities

## Abstract

**Simple Summary:**

Evidence on the use of brachytherapy in soft-tissue sarcomas (STSs) is sparse. Therapy regimens are determined more by local interdisciplinary tumor conferences than by standardized protocols. Patient-specific factors complicate the standardized application of therapy protocols. The individuality of the treatment makes it difficult to compare results. With a systematic review, we aim to provide the community dedicated to the treatment of soft-tissue sarcomas with a literary summary of the current literature.

**Abstract:**

Introduction: Evidence on the use of brachytherapy in soft-tissue sarcoma (STS) is sparse. Therapy regimens are determined more by local interdisciplinary tumor conferences than by standardized protocols. Patient-specific factors complicate the standardized application of therapy protocols. The individuality of the treatment makes it difficult to compare results. Materials and Methods: A comprehensive literature search was conducted, whereby the literature from a period of almost 44 years (1977–2021) was graded and included in this systematic review. For this purpose, PubMed was used as the primary database. Search string included “soft-tissue sarcoma”, “brachytherapy”, and “extremity.” Four independent researchers reviewed the literature. Only full-text articles written in English or German were included. Results: Of the 175 identified studies, 70 were eligible for analysis based on the inclusion and exclusion criteria. The key points to compare were local complications, recurrence rate and correlation with margins of resection, and the use of brachytherapy regarding tumor grading. Conclusion: Brachytherapy represents an important subset of radiotherapy techniques used in STSs, whose indications and applications are constantly evolving, and for which a local control rate of 50% to 96% has been reported as monotherapy, depending on risk factors. However, the best benefit is seen in the combination of further resection and brachytherapy, and most authors at many other centers agree with this treatment strategy.

## 1. Introduction

Soft-tissue sarcomas (STSs) are a heterogeneous and infrequent group of tumors consisting of approximately 1% of neoplasms diagnosed in the adult population and account for over 20% of all pediatric solid malignant cancers [1,2]. The experience collected in the past four decades has enhanced the crucial role of radiotherapy in the treatment of these diseases. To improve the local control of STSs, several radiotherapy techniques have been developed, one of which is brachytherapy (BRT). Radioactive elements in the tumor bed or directly introduced into the neoplasm allow radiation to be administered over a short distance [3]. This feature differentiates BRT from external-beam radiotherapy (EBRT), where photons or electrons are produced with a linear accelerator. A positive side effect is that the tumor or, after its excision, the tumor bed is irradiated directly; thus, the overlying layers and, because of its low penetration depth, also the deeper tissue structures are spared [4,5,6].

Nowadays, limb-sparing surgery associated or not with radiotherapy (RT) seems to be the gold standard treatment for STSs, achieving local control rates of approximately 85–90% and curative rates of 50% [1,7]. Histology, stage, primary localization, and resection margins are important factors influencing recurrence, which usually happens in the first 18 months after operation [8,9,10,11]. The possibility to augment surgical treatment and, therefore, the local control is the major characteristic of BRT in the treatment of extremity sarcomas, allowing limb-sparing surgery rather than salvage amputation [8,12]. 

Currently, brachytherapy can be used in three different forms: neoadjuvant, intraoperative, adjuvant, and as a separate treatment method for tumors that cannot be removed surgically [13]. In relation to surgery, BRT can have a neoadjuvant or adjuvant role in intraoperative radiotherapy (IORT) or postoperatively or could be used alone in cases of surgically untreatable STSs [6]. BRT and EBRT can also be used in cases of high-grade sarcomas [14]. The advantage of brachytherapy is the fact that it permits applicators to be inserted under visual control. Therefore, the process is very precise and diminishes the number of complications [13].

According to the time scale needed to deliver the radiation dose (dose rate), BRT is nowadays available as an HDR (high-dose rate), an LDR (low-dose rate), and, recently, an ultra-low-dose rate. HDR can be administered twice a day for seven days as fractionated HDR, to overcome patients’ confinement and prolonged shielding [3]. Technically, administration is possible through flaps or seeds in the surgical bed or into the tumor mass [3].

Most treatment protocols for extremity STSs use high doses of postoperative EBRT. In order to reduce the dose-related complications (such as fibrosis, loss of agility, limb edema, radiation dermatitis, and neuritis) BRT appears to be a potential alternative to EBRT without renouncing the positive effects of irradiation [15,16,17,18,19,20,21]. The selection of the target area, avoiding more healthy tissue, and a reduction in the length of RT additionally allow adjuvant therapy to be started earlier [1,22]. IORT is delivered using HDR-BRT with flexible applicators (“flaps”) on the tumor bed or with forward-directed electron beams [8,23,24].

Despite the positive data supporting IORT, brachytherapy is not a standard method used for the complementary treatment of soft-tissue sarcomas [13]. The biggest problem seems to be its limited accessibility and technical limitations resulting from the fact that brachytherapy facilities are rarely located near surgical or oncologic orthopedic departments offering the possibility of a shared surgical path [13,25]. The recent introduction of portable linear accelerators, delivering low-energy (50 kV) photons, could be an option for solving this limitation [8,26]. With regard to limb-sparing surgery and some metastases, the treatment of soft-tissue sarcomas and BRT evolves constantly. Metastases are, for example, found in the lungs, muscles, abdomen, and in regional lymph nodes. Regional lymph node metastases are usually rare (<3%), but there are exceptions such as epithelioid sarcomas, clear cell sarcomas, synovial sarcomas, and angiosarcomas [27]. This review aims to summarize the current role of brachytherapy in adults with soft-tissue sarcomas (STSs).

## 2. Material and Methods

A comprehensive literature search was conducted, and studies published between 1 January 1977 and 15 November 2021 were included in this systematic review. PubMed was used as the primary database for the literature search. Additional potentially matching studies were identified by cross-searching article references through a backward and forward citation search. The review was performed in accordance with the Preferred Reporting Items for Systematic Reviews and Meta-Analyses (PRISMA) guidelines [28] Therefore, PubMed’s literature was searched using the following search string: *soft-tissue sarcoma AND brachytherapy AND extremity.* Four authors independently screened the published studies by title and thereafter by the given abstract. Of these publications, all reviews, prospective and retrospective studies, and case reports were included. Furthermore, studies in English and German were included. Finally, only articles for which the full text was available were considered. Characteristics of all cases are shown in Table 1. The flowchart for the literature analysis is given in Figure 1.

## 3. Results

A total of 175 studies were identified, of which 70 were eligible for analysis based on the inclusion and exclusion criteria. They were analyzed according to (a) local complications, (b) the recurrence rate and its correlation with margins of resection, and (c) the use of BRT in regard to tumor grading. In 39 cases, the level of evidence (LoE) was 3, while in 17 published studies, the LoE was 5. In nine manuscripts, LoE was two; in four papers, it was four; and in one study, LoE was seven.

### 3.1. Local Complications

The most often described complications were wound complications (often classified as major, moderate, and minor complications), edema, fibrosis, bone loss, bone fractures, and peripheral nerve damage. Alekhteyar et al. compared BRT and BRT + EBRT and showed that there was no significant difference in terms of wound healing (*p* = 0.3), but with BRT alone, 26% of wound complications were observed [41]. Alektiar et al. demonstrated that there was no significant difference in wound complications with or without intraoperative BRT (*p* = 0.13), but the reoperating rate after wound complications was significantly higher (*p* = 0.02) in the BRT group [44]. In particular, the width of excised skin (WES) > 4 cm was associated with complications in the BRT group. They also concluded that the combination with radiotherapy is already a well-established expert opinion, but there is still disagreement about the timing and the procedure in terms of local complications and survival rate. “In terms of local control, the results with brachytherapy, preoperative radiation, or postoperative radiation seem to be similar…the dose rate of BRT and BRT dose to the skin had no significant impact on the wound reoperation rate” [44]. There was no increased incidence of peripheral nerve damage, but there was a significant increase in wound complications while using BRT before the fifth postoperative day (*p* = 0.05) [44]. In a non-randomized study, Alektiar et al. compared BRT versus IMRT (intensity-modulated radiotherapy) and did not find a statistically significant difference in wound complications [70]. In contrast, Arbeit et al. found a significantly increased number of major (and moderate) wound complications in comparison to a randomized group of patients, in which the BRT group had 22% wound complications versus 3% in the non-BRT group (*p* = 0.002) [31]. The study by Aronowitz et al. showed a relationship between the radiation dose and disturbed wound healing, where toxicity was correlated with the fraction size and total dose. The authors could not find a relationship with clinical or surgical factors. From their observations, they suggested the following dosing guidelines: The total SDD (source-detector distance) should not be greater than 15 Gy, delivered in (three or four) fractions of <450 cGy [59]. No one clearly addressed the question of whether different tumor entities or localization of the body region might have an impact on the local complication rate. Regarding the use of brachytherapy, there are arguments for and against this approach. Pre-, intra-, and postoperative use favors local complications: Preoperative use doubles the risk of wound complications, whereas postoperative use increases the probability of late side effects: edema, fibrosis, and joint stiffness [68,77].

Overall, it can be stated that radiation-induced toxicity depends on the dose and the volume treated. BRT can spare normal tissue better than can EBRT because radiation can be directed into the surgical bed while minimizing radiation to healthy tissue [4,5,6,67]. Nevertheless, influenced by multiple factors, acute as well as chronic complications occur in the treatment of STSs, with delayed wound healing being the most frequent [4,44,45,81,83]. Consequently, the entry point of the catheters should be at least 1–1.5 cm from the wound edge [4,44,45]. In the case of postoperative radiotherapy, radiation delivery begins after wound closure, which may make it difficult to protect normal tissue. Therefore, tissue expanders with removable (e.g., drains) or absorbable materials (mesh, gel) may be applied to protect critical structures in the postoperative period. In addition, the use of temporary closure (i.e., negative pressure wound therapy (NPWT) or synthetic dressing) can minimize the radiation dose to the edges of the healing wound, which may reduce acute toxicity, thus resulting in fewer wound complications and less dehiscence or edema [78]. Continuous pressure of −125 mmHg is applied during NPWT. In the treatment of sarcomas, NPWT has been shown to reduce the rate of wound complications after surgery, decrease the extent of wound closure required, and reduce the dose applied to normal tissue during brachytherapy [78]. To minimize catheter movement, NPWT should be replaced only after the completion of BRT [4]. To avoid wound complications, radiotherapy should be avoided until final wound closure or, if necessary, should be started on postoperative day 5 at the earliest [4,6,44]. The method of closure also has to be considered, with free tissue transfer being a good option when brachytherapy treatment is required due to the reduction in wound tension. In cases with a high anticipated risk of wound complications, closure with fresh vascularized tissue may be an option [44]. These risk factors include wound diameter (>100 cm^2^), excised skin width (>4 cm), target volume (>210 cm^3^), lower extremity disease (i.e., popliteal/posterior thigh), major neurovascular tumor involvement, previous resection/radiation, or medical comorbidities (i.e., smoking, diabetes, vascular disease, etc.) [4]. Overall, the complication rate reported in different publications ranges between 5% and 75%. These adverse effects concern peripheral nerves, less frequently, the skin, and very rarely, radiation-induced muscle damage and bone loss [45,81].

### 3.2. Recurrence Rate and Correlation with Margins of Resection

Alekhteyar et al. compared BRT alone and BRT + EBRT, where no significant difference (*p* = 0.32) could be seen with an overall two-year local control rate of 86% [41]. However, they identified two factors in relation to the local recurrence rate: (a) primary tumor and (b) resection margins. In detail, a primary tumor and negative margins of resection were in patients with high-grade soft-tissue sarcomas of the extremities associated with improved local control compared with a recurrent tumor and positive margins of resection. Even if their sample size did not show significance, with a positive resection margin with BRT + EBRT, they observed 90% local control versus 59% with BRT alone (*p* = 0.08) [41]. They treated patients who had been resected R0 solely with BRT and thus achieved a local control rate of 94% but suggested that an additional EBRT should be considered if the resection margin is positive. In a non-randomized study, Alektiar et al. examined whether BRT or IMRT showed better local control and concluded that although the IMRT group had more patients with positive margins and larger tumors, the group had a local control rate of 92% vs. 81% in the BRT group (*p* = 0.04). Therefore, they suggested a further examination of primary treatment in STSs [70]. Arbeit et al. reported a recurrence rate of 20–60 % for single-use surgical treatment [31]. The recurrence occurred most often within the first two years after primary surgery [71]. The extent of the resection margin was found to positively influence disease-free survival (*p* = 0.002) and disease-specific survival (*p* = 0.002) [71]. Tumor size was related to disease-specific survival [71].

Most authors stated that surgical procedures alone can achieve good local control, i.e., a recurrence-free state in two-thirds of cases. However, some authors have highlighted the role of clinical features such as a size of less than 4 cm, low grade, and an epifascial location. Tumor size is an essential factor [68,77]. The larger the tumor, the greater the displacement of the surrounding tissue, and the greater the postresection substance defect. In contrast, these authors showed that in one group, local control could be achieved in about 50% of the patients with soft-tissue sarcomas smaller than 10 cm using definitive radiotherapy (63 Gy) without surgery [68,77]. Patients with low-grade tumors who did not receive chemotherapy were randomized in an observational study with the result that at a median follow-up time of 9.6 years, patients who were solely treated with limb-sparing surgery had an increased local recurrence rate (24.3% vs. 1.4%). Thus, it appears that BRT increases local control in addition to limb-sparing surgery [68,77]. Brachytherapy has been used as an adjuvant treatment strategy to reduce the risk of local recurrence [87]. However, subsequent failures have been reported in more than 50% of such cases [88]. Thus, brachytherapy in STSs can significantly decrease the number of local recurrences, whereas it has no direct effect on the number of distant metastases [88].

### 3.3. Brachytherapy and Tumor Grading

Before starting therapy, all patients should be investigated and managed by a multidisciplinary team with experience in the field of sarcomas [68,77]. Using a synopsis of clinical and radiological findings, the tumor board establishes a working diagnosis, as well as a diagnostic chain, in which an ultrasound- or CT-targeted biopsy (in individual cases, also an excisional biopsy) provides an intralesional tissue sample. After histological–pathological processing, diagnosis and, accordingly, staging and grading are performed. If a surgical procedure is used in the therapy, the material obtained there is comparatively re-examined [4]. In addition to size, tumor grade is a well-established risk factor for local or systemic tumor recurrence, leading to different treatment modalities [5]. Limb-sparing surgery without radiotherapy may be the optimal local therapy for extremity soft-tissue sarcomas that are small (<5 cm), low-grade tumors, and superficial to the fascia [4,56,77]. In cases of high-grade STSs, additional RT in the form of brachytherapy or external beam radiation is well recognized and remains the standard therapy following limb-sparing surgery [4,84,85]. Harrison et al. of the Memorial Sloan Kettering Cancer Center in New York presented a prospective, randomized study in 1993, in which the addition of brachytherapy in the treatment of high-grade STSs provided a benefit in local control, compared with surgery alone (90% vs. 65%, *p* = 0.013) [37]. Pistors et al. confirmed these findings in a larger cohort with a median follow-up of 76 months, in which patients were randomized to receive either no additional therapy or brachytherapy [14]. For brachytherapy, catheters were placed intraoperatively in the tumor bed and loaded five days later with low-dose (LDR) 192-iridium for four to six days to deliver 42 Gy to 45 Gy, resulting in a local control rate for high-grade soft-tissue sarcomas of 89% with BRT and 66% without BRT (*p* = 0.0025) [14]. However, overall survival was not affected in either study [14,37,77]. Small tumor size and a short time interval between radiotherapy and surgery also seem to improve the outcome [71].

## 4. Discussion

A problem encountered when working through the studies was the lack of comparability. The reason lies in the fact that various authors have compared different methods against each other. Amputation, limb-saving surgery, and the use of radiotherapy are three procedures, among which radiotherapy can additionally be administered at three different times. It is already clear that, in terms of effect, comparability is very difficult. Prospective, randomized trials comparing EBRT and BRT were not found, possibly because of some BRT-limiting factors, such as technical demands and lack of effectiveness in low-grade histology and special tumor-bed geometry [89]. The optimal form and time of adjuvant radiation are unclear, and most studies on the use of radiation in the treatment of sarcomas are based on EBRT [45]. Brachytherapy represents an important subset of radiotherapy techniques used in STSs, whose indications and applications are constantly evolving and for which a local control rate in the range of 50% to 96% has been reported as monotherapy, depending on risk factors [1]. A short time interval between radiotherapy and surgery also seems to improve the outcome [71]. Authors agree that surgical treatment in STSs is the gold standard; the question is with which type of radiation over what time should we combine limb-sparing surgery and the different types of radiation. The factors that seem to be associated with better situations are primary tumor, a negative margin of resection, WES < 4 cm, and the use of BRT after postoperative day 5 [41,45,70,89]. Furthermore, shoulder tumors are associated with a less favorable outcome [70]. In addition, several studies agree that RT can improve local control in patients with close margins. In patients undergoing RT and limb-sparing surgery for STSs, achieving a negative margin is essential for optimizing local control as well as for overall survival [41,45,70,89]. Therefore, the surgical resection of the tumor with a safety margin of healthy tissue is the fundamental treatment method for soft-tissue sarcomas. However, the administration of brachytherapy appears to be associated with improved local control and a lower rate of recurrence [90]. Thus, brachytherapy in STSs can significantly decrease the number of local recurrences, whereas it has no direct effect on the number of distant metastases [13]. However, the absolute quantitative width of the negative margin does not significantly influence the outcome; thus, attempts at wide margins of resection appear to be unnecessary [91].

Arbeit et al. observed a significantly higher rate of major wound complications in the brachytherapy group than in the group with no brachytherapy (22% vs. 3%, *p* = 0.002) [31]. This stands in contrast to the results published by Alektiar et al., who found no significant increase in wound complications between BRT monotherapy and surgery alone in a randomized trial (24% vs. 14%, *p* = 0.13) [44]. However, a research group from the same hospital observed significant wound complication rates of 48% when BRT started within the first five postoperative days. The wound complication rate dropped when BRT started earliest from postoperative day 5 (17% BRT vs. 15% for surgery alone, *p* = 0.9) [32,44]. Furthermore, adding EBRT to BRT seems not to significantly affect the overall wound complication rate (26% BRT vs. 38% BRT + EBRT, *p* = 0.31) [41]. Nevertheless, the optimal form and time of adjuvant radiation are unclear.

A positive side effect observed by Alektiar et al. was that patients treated with BRT would leave the hospital within two weeks after having completed all treatment, whereas patients with EBRT required six to seven weeks of treatment [41]. There seems to be a time-dependent effect of radiation on wound complications. In general, when it comes to local complications, these adverse effects concern peripheral nerves, less frequently, the skin, and very rarely, radiation-induced muscle damage and bone loss with consecutive fractures. Overall, the complication rate reported in different publications ranged between 5% and 75%. Peripheral nerve damage was the major adverse effect, comprising about 5% [25,45,81]. When different adjuvant treatment options were compared, some studies reported a higher rate of deep infection in patients treated with high-dose brachytherapy and surgical debridement than those treated with EBRT. In contrast, a higher risk of late side effects was reported for patients treated with EBRT and surgical resection, including chronic edema, fibrosis, chronic radiation dermatitis, and fracture [72]. However, it seems that the incidence of acute complications does not translate to substantial long-term morbidity following brachytherapy [81]. The surgical resection of the tumor with a safety margin of healthy tissue is the fundamental method in the treatment of soft-tissue sarcomas. However, in large soft-tissue sarcomas or sarcomas that cannot be completely surgically resected, therapy should include a combination of surgical intervention and radiotherapy. In our opinion, brachytherapy is preferable, when possible. Adjuvant brachytherapy increases the local control rate to up to 78%, is well tolerated, and rarely causes complications. Treatment should be delivered in specialist centers with multidisciplinary resources [13,92].

## 5. Conclusions

Surgical excision is the gold standard in the treatment of soft-tissue sarcomas of the extremities. During the review of the literature, the authors identified factors that were associated with a favorable prognosis, including primary tumor, a negative margin of resection, WES < 4 cm, and the use of BRT after postoperative day 5. By contrast, the following factors were associated with unfavorable outcomes: recurrent tumor, a positive margin of resection, WES > 4 cm, the use of BRT before postoperative day 5, and shoulder tumor.

However, brachytherapy appears to be a good additive method of adjuvant therapy. It provides a benefit in local control, compared with surgery alone. In connection with improved local control and a lower rate of recurrence, there is no statistical significance in overall survival. In general, the number of complications appears to be small and concerns mostly peripheral nerves, less frequently, the skin, and very rarely, radiation-induced muscle damage and bone loss. There is still no consensus on which is the best form of adjuvant treatment. Further multicenter randomized studies are necessary to decide which is the best method of adjuvant treatment, and what is the optimal treatment for improving local control.

## Figures and Tables

**Figure 1 cancers-15-01133-f001:**
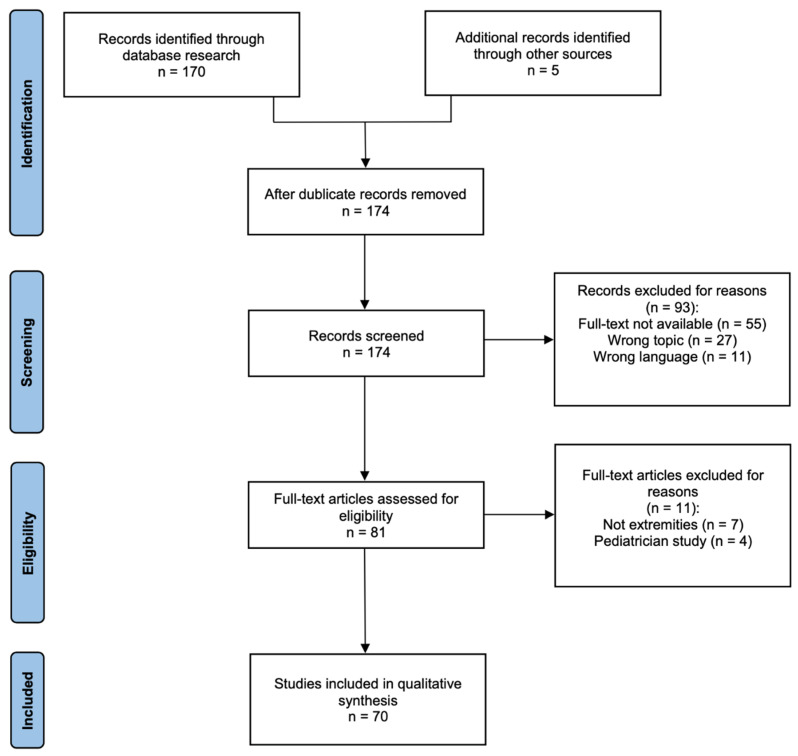
Flowchart of literature research and analysis.

**Table 1 cancers-15-01133-t001:** Detailed study characteristics of included publications with level of evidence (LoE).

ID	Study	Year	Region	Country	Sample Size	Follow-Up	Treatment	Study Type	LoE
1	Mills et al. [29]	1981	Africa	South Africa	17	28 months	HD-BRT	Retrospective study	3
2	Brennan et al. [30]	1987	North America	USA	117	16 months	BRT vs. No BRT	Prospective randomized trial	2
3	Arbeit et al. [31]	1987	North America	USA	105	11.9 months	BRT vs. No BRT	Prospective randomized trial	2
4	Ormsby et al. [32]	1989	North America	USA	52	3 months	BRT vs. No BRT	Retrospective study	3
5	Zelefsky et al. [33]	1990	North America	USA	45	4 years	BRT	Retrospective study	3
6	Nori et al. [34]	1991	North America	USA	40	36 months	BRT	Retrospective study	3
7	Brennan et al. [35]	1991	North America	USA	126	40.8 months	BRT vs. No BRT	Prospective randomized trial	2
8	Habrand et al. [36]	1991	Europe	France	48	82 months	BRT	Retrospective study	3
9	Harrison et al. [37]	1993	North America	USA	126	66.5 months	BRT vs. No BRT	Prospective randomized trial	2
10	Pisters et al. [38]	1994	North America	USA	45	67 months	BRT vs. No BRT	Prospective randomized trial	2
11	Janjan et al. [39]	1994	North America	USA	35	n.a.	BRT vs. EBRT	Comparative study	3
12	Catton et al. [40]	1996	North America	Canada	25	24 months	BRT or EBRT or BRT + EBRT vs. Surgery alone	Retrospective study	3
13	Alekhteyar et al. [41]	1996	North America	USA	105	22 months	BRT vs. BRT + EBRT	Retrospective study	3
14	Pisters et al. [14]	1996	North America	USA	164	76 months	BRT vs. No BRT	Prospective randomized trial	2
15	Panchal et al. [42]	1996	Europe	United Kingdom	4	27.5 months	Surgery + BRT	Retrospective study	3
16	Chaudhary et al. [43]	1998	Asia	India	151	24 months	BRT vs. BRT + EBRT	Comparative study	3
17	Alektiar et al. [44]	2000	North America	USA	164	100 months	BRT vs. No BRT	Prospective randomized trial	2
18	Alektiar et al. [45]	2002	North America	USA	202	61 months	BRT	Retrospective study	3
19	Mccarter et al. [46]	2002	North America	USA	n.a.	n.a.	n.a.	Review	5
20	Ballo et al. [3]	2003	North America	USA	n.a.	n.a.	n.a.	Review	5
21	Rachbauer et al. [47]	2003	Europe	Austria	39	26 months	HD-BRT + EBRT	Prospective study	2
22	Strander et al. [48]	2003	Europe	Sweden	n.a.	n.a.	n.a.	Review	5
23	Murray et al. [49]	2004	North America	USA	n.a.	n.a.	n.a.	Review	5
24	Maples et al. [50]	2004	North America	USA	n.a.	n.a.	n.a.	Review	5
25	Kretzler et al. [51]	2004	Europe	Germany	28	4.3 years	BRT ± EBRT	Retrospective study	3
26	Fontanesi et al. [52]	2004	North America	USA	31	60.5 months	Surgery ± BRT ± EBRT	Retrospective study	3
27	Baumert et al. [53]	2004	Europe	Switzerland	1	n.a.	BRT	Case report	4
28	Moureau-Zabotto et al. [54]	2004	Europe	France	83	13 years	Surgery ± BRT ± EBRT	Retrospective study	3
29	Fontanesi et al. [55]	2004	North America	USA	13	76 months	Surgery ± BRT ± EBRT	Retrospective study	3
30	Schuetze et al. [56]	2005	North America	USA	n.a	n.a.	n.a.	Review	5
31	DeLaney et al. [57]	2005	North America	USA	n.a	n.a.	n.a.	Review	5
32	Martínez-Monge et al. [25]	2005	Europe	Spain	25	23.2 months	HD-BRT + EBRT	Retrospective study	3
33	Lazzaro et al. [58]	2005	Europe	Italy	42	34 months	BRT ± EBRT	Retrospective study	3
34	Aronowitz et al. [59]	2006	North America	USA	12	34 months	HD-BRT	Retrospective study	3
35	Mierzwa et al. [60]	2007	North America	USA	43	39 months	BRT ± EBRT	Retrospective study	3
36	Torres et al. [61]	2007	North America	USA	62	6 years	BRT vs. No BRT	Retrospective study	3
37	Laskar et al. [20]	2007	Asia	India	155	45 months	BRT ± EBRT	Retrospective study	3
38	Pohar et al. [62]	2007	North America	USA	37	47 vs. 17 months	LD-BRT + EBRT vs. HD-BRT + EBRT	Retrospective study	3
39	Beltrami et al. [5]	2008	Europe	Italy	112	75 months	BRT + EBRT	Retrospective study	3
40	Muhic et al. [63]	2008	Europe	Denmark	39	3.4 years	PDR-BRT + EBRT	Retrospective study	3
41	Kaushal et al. [64]	2008	North America	USA	n.a.	n.a.	n.a.	Review	5
42	Rimner et al. [65]	2009	North America	USA	255	71 months	BRT or EBRT or BRT + EBRT	Retrospective study	3
43	Rudert et al. [66]	2009	Europe	Germany	n.a.	n.a.	n.a.	Review	5
44	Petera et al. [67]	2010	Europe	Czech Republic	45	3.2 years	BRT ± EBRT	Retrospective study	3
45	Shukla et al. [68]	2011	Asia	India	300	n.a.	BRT ± EBRT	Retrospective study	3
46	Bradley et al. [69]	2011	North America	USA	11	20.8 months	HD-BRT	Retrospective study	3
47	Alektiar et al. [70]	2011	North America	USA	134	46 months	LD-BRT or IMRT	Retrospective study	3
48	Atean et al. [71]	2012	Europe	France	87	69 months	EBRT vs. EBRT + BRT	Retrospective study	3
49	Guzik et al. [13]	2012	Europe	Poland	1	n.a.	BRT	Case report	4
50	Emory et al. [72]	2012	North America	USA	190	40 months	EBRT or BRT or BRT + EBRT	Retrospective study	3
51	Delaney et al. [73]	2012	North America	USA	n.a.	n.a.	n.a.	Review	5
52	Ghadimi et al. [74]	2014	Europe	Germany	n.a.	n.a.	n.a.	Review	5
53	Pellizzon et al. [6]	2014	South America	Brazil	n.a.	n.a.	n.a.	Review	5
54	Ren et al. [15]	2014	Asia	China	110	43.7 months	BRT	Retrospective study	3
55	Miller et al. [75]	2015	North America	USA	n.a.	n.a.	n.a.	Review	5
56	Röper et al. [76]	2015	Europe	Germany	n.a	n.a	n.a	Prospective study	3
57	Larrier et al. [77]	2016	North America	USA	n.a.	n.a	n.a	Review	5
58	Naghavi et al. [78]	2016	North America	USA	40	27 months	BRT	Retrospective study	3
59	Mukherji et al. [79]	2017	Asia	India	3	34 months	BRT	Case report	4
60	Naghavi et al. [4]	2017	North America	USA	n.a.	n.a.	n.a.	Review	5
61	Cortesi et al. [80]	2017	Europe	Italy	107	100 months	BRT + EBRT	Retrospective study	3
62	Correa et al. [1]	2018	Europe	Spain	n.a.	n.a.	n.a.	Review	5
63	Klein et al. [81]	2018	North America	USA	171	71.8 months	HD-BRT or EBRT or HD-BRT + EBRT	Retrospective study	3
64	Healey et al. [82]	2018	North America	USA	n.a.	n.a.	n.a.	Expert opinion	7
65	Manir et al. [24]	2018	Asia	India	27	20 months	BRT ± EBRT	Retrospective study	3
66	Gimeno et al. [83]	2019	Europe	Spain	106	7.1 years	HD-BRT + EBRT	Prospective controlled study	2
67	Spoto et al. [84]	2020	Europe	Italy	90	4.2 years	BRT vs. EBRT vs. BRT + EBRT	Retrospective study	3
68	Roeder et al. [85]	2020	Europe	Austria	n.a.	n.a.	n.a.	Review	5
69	Sarria et al. [8]	2020	Europe	Germany	31	4.9 years	BRT	Retrospective study	3
70	Vavassori et al. [86]	2021	Europe	Italy	1	40 months	HD-BRT	Case report	4

BRT, brachytherapy; HD, high dose; LD, low dose; PDR, pulsed dose rate; EBRT, external beam radiotherapy; IMRT, intensity-modulated radiotherapy; n.a., not available.

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
