# Peer review of "Brachytherapy in the Treatment of Soft-Tissue Sarcomas of the Extremities—A Current Concept and Systematic Review of the Literature"

_cancers, 2023, doi:10.3390/cancers15041133_

Round 1

Reviewer 1 Report

I think this is a well written review article.  I have no significant changes I think need to be made.

Reviewer 2 Report

Neugebauer et al. provide a thorough overview of the role and use of brachytherapy in the treatment of soft tissue sarcoma of the extremities. Their work reviews all relevant literature available and provides some conclusions to be integrated into clinical practice.

However, I have several remarks:

The work could be written more fluently. Consider having it read and corrected by a native English speaker.

 Title: Consider adding the term "extremity" to the title.

  Introduction: The last sentence of the first paragraph is unclear, consider rephrasing. 59) Radiation dermatitis should be added as a (most frequent) side effect of extremity sarcoma EBRT (to also underscore the differences with BT). 71) What is meant with "some metastases"?

 Methods: Consider ranking the papers in Table 1 based on their year of publication. It would be of great added value to add sample size and follow-up time to Table 1, where available, to get a better sense of the validity of the evidence presented. Furthermore, a short summary of what was tested could be useful (e.g. "BT vs. BT+EBRT").

 Results: The study without LoE should be classified as LoE VII (expert opinion), as it is just a comment. 128) What is meant with "favors local complications"? Consider rephrasing. 162) Provide information on the primary tumor, i.e. which histologies have best/worst outcome? EBRT in STS notoriously makes use of very wide safety margins to control for microscopic disease (e.g. up to 4-5 cm depending on institutional standards): Is anything mentioned about this difference in treated volume in comparison with BT?

 Discussion: 279) Parts of this paragraph are verbatim repetitions of other sentences in the discussion. Consider rephrasing and omitting double sentences to increase fluency.

 Conclusion: Table 1 should be renamed to Table 2.

Reviewer 3 Report

Title:

Brachytherapy in the Treatment of Soft Tissue Sarcomas - A Current Concept and Systematic Review of the Literature.

Concise Summary:

Authors aimed to review the role of brachytherapy in Soft Tissue Sarcomas (STS). The key points to evaluate were local complications, recurrence rate and the use of brachytherapy regarding tumor grading. It is concluded that brachytherapy improved local control rate, but not overall survival, and without serious side effects.

Major Comments:

It is an interesting review, although the contribution on the issue is relatively scarce and lack of originality.  It is largely known that limb-sparing surgery associated or not with radiotherapy is the current gold standard treatment for STS. Brachytherapy is not a standard method used for treatment of STS, but the effectivity of this treatment to local control of STS is also known, although it produces obviously side effects.

The structure of the text in Results and Discussion is insufficiently substantiated given it entails unnecessary data repetition.

It is noteworthy that the authors do not provide any publications in this regard or any experience in Radiotherapy or Pathology, so their contribution seems to be solely to evaluate what is written in the literature. It seems a relevant disadvantage when it is necessary to provide, in addition to the review, a accredited critical opinion. Furthermore, there is no proven affiliation of the authors with Radiotherapy or Pathology to give the text a comprehensive view.

It is said (Line 196) that “In the diagnostic workup, a tissue biopsy is required to confirm the stage, histology and tumor grade [4].”  The pathological study does not confirm any diagnostic, because it gives the diagnostic, independently of the previous clinic-radiological approach. Furthermore, obviously the tissue biopsy does not confirm any diagnosis. This way of expressing the idea is wrong. In addition, the reference is unnecessary.

Finally, it is not acceptable to introduce a table in Conclusion section.

Conclusion:

The article addresses an interesting clinical issue. However, I consider that the text should be improved by giving it more quality and originality. Therefore, I think that this article has not enough quality to be published in the Journal

Round 2

Reviewer 2 Report

Dear authors, thank you for considering my remarks. Especially the extension of Table 1 makes for a big improvement of your work.

To clarify your remaining questions:

- Radiation dermatitis should be added on line 69 of the new version.- Line 81: I suggest writing "some metastases" and then specifying with a couple of examples.- Line 174: a) primary tumor: please specify what Alekhteyer et al. wrote (which body regions are meant ? which histologies ?)

Good luck with your publication.

Reviewer 3 Report

I consider that it is an interesting article, that provides un update about both the role of brachytherapy in Soft Tissue Sarcomas (STS). Although the information provided by the review is useful, it is still incomplete and provisional. The authors have correctly argued the reviewer's observations and criticisms. Taken together the advantages and disadvantages of the proposed review, since the article that it can give researchers a more detailed information than it is available in the current literature, I think this article has enough quality to be accepted for publication.

However, the authors’ answer to the reviewer's observations about “pathological study does not confirm any diagnostic, because it gives the diagnostic, independently of the previous clinical-radiological approach” has not been understood. Of course, we know patient cases are discussed in the Tumor Board to achieve both the best diagnosis and treatment, but it can not be stated that the pathological diagnosis “confirm” the diagnosis, because it “gives” the diagnosis. Consequently, the authors statement “In the diagnostic workup, a tissue biopsy is required to confirm the stage, histology and tumor grade [4].”  should be modified accordingly
